# Text-Region Matching for Multi-Label Image Recognition with Missing Labels

### Leilei Ma
xiaomylei@163.com
School of Computer Science and
Technology
Anhui University
Hefei, Anhui, China

### Hongxing Xie
xiehongxing2001@163.com
School of Artificial Intelligence
Anhui University
Hefei, Anhui, China

### Lei Wang
lei_wang@njust.edu.cn
School of Computer Science and
Engineering
Nanjing University of Science and
Technology
Nanjing, Jiangsu, China

### Yanping Fu
ypfu@ahu.edu.cn
School of Computer Science and
Technology
Anhui University
Hefei, Anhui, China

### Dengdi Sun*
sundengdi@163.com
School of Artificial Intelligence
Anhui University
Hefei, Anhui, China

### Haifeng Zhao*
senith@163.com
School of Computer Science and
Technology
Anhui University
Hefei, Anhui, China

## Abstract

Recently, large-scale visual language pre-trained (VLP) models have demonstrated impressive performance across various downstream tasks. Motivated by these advancements, pioneering efforts have emerged in multi-label image recognition with missing labels, leveraging VLP prompt-tuning technology. However, they usually cannot match text and vision features well, due to complicated semantics gaps and missing labels in a multi-label image. To tackle this challenge, we propose **T**ext-**R**egion **M**atching for optimizing **M**ulti-**L**abel prompt tuning, namely TRM-ML, a novel method for enhancing meaningful cross-modal matching. Compared to existing methods, we advocate exploring the information of category-aware regions rather than the entire image or pixels, which contributes to bridging the semantic gap between textual and visual representations in a one-to-one matching manner. Concurrently, we further introduce multimodal contrastive learning to narrow the semantic gap between textual and visual modalities and establish intra-class and inter-class relationships. Additionally, to deal with missing labels, we propose a multimodal category prototype that leverages intra- and inter-category semantic relationships to estimate unknown labels, facilitating pseudo-label generation. Extensive experiments on the MS-COCO, PASCAL VOC, Visual Genome, NUS-WIDE, and CUB-200-211 benchmark datasets demonstrate that our proposed framework outperforms the state-of-the-art methods by a significant margin. Our code is available here ○ .

*Corresponding Authors: Dengdi Sun and Haifeng Zhao. All except Lei Wang are members of the Anhui Provincial Key Laboratory of Multimodal Cognitive Computation.

## CCS Concepts

• **Computing methodologies** → Object recognition.

## Keywords

Multi-Label Image Recognition, Vision and Language, Prompt-Tuning, Text-Region Matching

**ACM Reference Format:**
Leilei Ma, Hongxing Xie, Lei Wang, Yanping Fu, Dengdi Sun, and Haifeng Zhao. 2024. Text-Region Matching for Multi-Label Image Recognition with Missing Labels. In *Proceedings of the 32nd ACM International Conference on Multimedia (MM '24), October 28-November 1, 2024, Melbourne, VIC, Australia.* ACM, New York, NY, USA, 10 pages. https://doi.org/10.1145/3664647.3680815

## 1 Introduction

Collecting high-quality and complete annotations for each image is time-consuming and labor-intensive, which hinders the application and promotion of multi-label image learning [11, 46]. An emerging research interest is relaxing the full-supervised setting to multi-label image recognition with missing labels (MLR-ML), meaning that only a subset of full labels is annotated in a multi-label image. MLR-ML has become increasingly popular owing to its reduced labeling costs, rendering it more practical for extensive use. However, this shift has also resulted in previous multi-label image recognition methods losing working (*i.e.*, incorrect utilization of incomplete annotated data or misuse), introducing new challenge.

To tackle this problem, some researchers [1, 13] have explored a few flexible solutions to study the task, such as utilizing known labels or assuming unknown labels as negative ones. In addition, Graph-based methods [6, 32] propose to model the correlation between images and labels to solve the MLR-ML problem through the semantic transfer or representation blending. Although they achieved acceptable results, their performance is significantly inferior when compared to prompt tuning based methods. For example, DualCoOp [36] pioneers the application of CoOp [55] to the MLR-ML task and achieves favorable performance, indicating that prompt

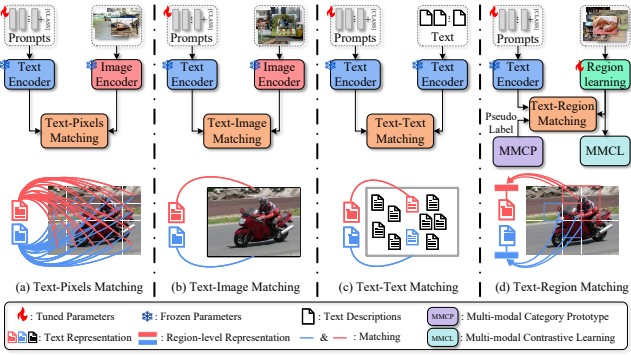

**Figure 1: Illustrate different matching methods for textual and visual representations, including text-pixel, text-image, text-text, and our proposed text-region matching. Unlike other methods, our approach establishes a one-to-one correspondence between textual and region visual representations.**

tuning on the CLIP [33] is a promising avenue for advancing MLR-ML. Additionally, TaI-DPT [16] and LLM-PT [49] advocate texts as images to augment training examples, while SCPNet [12] explores structured prior knowledge to learn label relationships.

*Nevertheless, these prompt tuning-based methods still exhibit limitations*: **1)** In terms of how text and visual representations match, SCPNet focuses on *text-image* matching, neglecting the rich semantic information within visual data. DualCoOp utilizes *text-pixels* matching, which can introduce irrelevant noise due to its emphasis on pixel-level details. As shown in Figure 1(a), there are more negatively correlated pixel-level representations than positively correlated ones for arbitrary textual representations under *text-pixels* matching. Regarding *text-image* matching in Figure 1(b) (*e.g.*, CoOp, SCPNet), image-level representations encompass multiple semantic objects and corresponding scenes, making it challenging for textual representations to distinguish different visual concepts. As a result, both *text-image* and *text-pixel* matching methods struggle to efficiently match the textual representation with its corresponding visual counterpart. Moreover, as described in Figure 1(c), *text-text* matching (*e.g.*, Tai-DPT, LLM-PT [49] and TaI-Adapter [56]) is susceptible to underperforming on test images because of the heterogeneity between training and testing data. **2)** Current prompt tuning methods [12, 16, 36, 49, 56] struggle to effectively leverage known annotation information, leading to discarding valuable data on unknown annotations. These methods often mask out unknown labels during the calculation of binary cross-entropy loss, hindering the model's ability to learn accurate label correlations. Specifically, taking an image with categories "*person*", "*dog*", and "*frisbee*" as an instance, when the category "*dog*" is unlabeled, the model will fail to build the correlation between "*dog*" and any of {"*person*", "*frisbee*"} during training, which hinders recognition of "*dog*" in inference. Facing extremely missing labels, *e.g.*, single positive label, the performance of existing methods will further decrease dramatically. **3)** Within the joint embedding space, existing prompt tuning based methods [12, 15, 16, 36] fail to effectively align visual and text representations. This results in a

significant semantic gap between modalities, hindering text-vision matching. Furthermore, these methods neglect both intra-class and inter-class semantic relationships within each modality, as well as the semantic relationships between modalities themselves. Although SCPNet [12] acknowledges the inter-class relationship within the text modality, it overlooks the intra- and inter-class relationships among visual representations and the interplay between visual and textual modalities.

Based on the above investigations, as shown in Figure 1(d), we design text-region matching for optimizing multi-label prompt tuning, which is collaborative with multimodal category prototype and multimodal contrastive learning. **Firstly**, we propose a Category-Aware Region Learning Module based on cross-modal attention to learn semantically relevant region-level visual representations corresponding to textual descriptions. In this way, one-to-many (*text-pixels*) or one-to-agnostic matching (*text-image*) is converted to one-to-one (*text-region*) matching. **Secondly**, visual representations are generally highly stochastic due to various contexts with external objects and object representations. Therefore, we represent each category with multiple visual prototypes to describe the object. Besides, we construct a corresponding text prototype for each category to leverage the textual knowledge embedded in CLIP effectively. Then, Multimodal Category Prototypes estimate unknown labels by leveraging intra- and inter-class semantic relationships within and across modalities. **Finally**, to bridge the semantic gap between vision and text and establish intra- and inter-class relationships, we propose Multimodal Contrastive Learning that maximizes intra-class similarity while minimizing inter-class similarity. This idea leverages the potential advantages of contrastive learning in learning representations [5, 42].

Our contributions to this work are summarized as follows:

1) The text-region matching strategy is proposed to narrow the impact of irrelevant visual information and the semantic gap between text and vision, which is a major contribution of this work.

2) Multimodal Category Prototype is designed to estimate unknown labels and generates pseudo-labels via intra- and inter-category semantic relationships.

3) Multimodal Contrastive Learning is introduced to further align the models between textual and visual representations. Besides, implicit inter-class relationships can also be learned to obtain fine visual representations.

4) We demonstrate through mathematical analysis how text-region matching is effective. The detailed mathematical analysis is presented in the appendix due to limited space.

## 2 Related Work

**Multi-label Image Recognition.** Multi-label image recognition (MLIR) [40, 48] is a crucial task in computer vision that involves assigning corresponding labels to various objects or semantic content contained in an image. As far as methods are concerned, they can be summarized into three categories: 1) *modeling label relationship* [8, 39, 43, 45]; 2) *object localization or attention mechanism* [7, 50, 51]; 3) *improving loss function* [34, 44]. Given the substantial cost of acquiring complete annotations, multi-label image recognition with missing labels tasks have attracted research attention. A naïve solution treats unknown labels as negative since

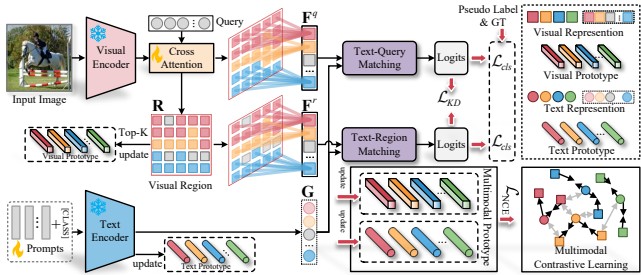

**Figure 2: The overview of the TRM-ML framework. We freeze the visual encoder and text encoder during training phase, and only allow the category query, cross-attention, and a simple MLP to be trained.**

there are more positive than negative labels per image [21, 35]. Nevertheless, these methods mistake positive labels as negative ones, resulting in undesired results. Inversely, some researchers only exploit known labels or generate pseudo-labels. For example, Durand *et al.* [13] propose a partial-BCE that only considers known labels and ignores unknown labels. Kim *et al.* [23] modify the label based on the largest loss value and use the changed label for the next training. Ben-Baruch *et al.* [1] design a partial asymmetric loss to select an appropriate threshold for each class to generate pseudo-labels by estimating the class distribution. Later, some researchers employ relational modeling to solve missing labels problems. SST [6] considers the structured semantic relationship, within and cross-images, to complete unknown label information. SARB [32] blends different label-level semantic representations to generate new labels and representations, then models the relationship between new representations on a co-occurrence graph.

**Prompt Tuning in Visual Tasks.** Visual prompt tuning comes from natural language processing (NLP) [27], a parameter-efficient learning method that is flexibly and efficiently adapted to various downstream vision tasks. CoOp [55] and its extension research [54] introduce content-optimizable prompt learning, which can perform satisfactorily with a few labeled images for each category. However, optimizing and utilizing text effectively descriptions is still a challenging problem. To alleviate this problem, ProDA [28] and PPL [25] aim to optimize the probability distribution of multiple prompts for the same visual representation using Gaussian Mixture Models. Besides, some prompt tuning-based methods are devoted to solving the multi-label recognition problem, *e.g.* DualCoOp [36], TaI-DPT [16], SCPNet [12], HSPNet [38], T2I-PAL [15], LLM-PT [49] and TaI-Adapter [56]. However, unlike the impractical text and visual matching approaches of the seven methods mentioned above, we optimize matching strategies to achieve one-to-one matching between text and corresponding visual regions.

**Contrastive Learning.** In recent years, contrastive learning [30, 41] has significantly advanced the domain of computer vision and steadily gained prominence as a mainstream technique. This methodology endeavors to learn discriminative representations by aligning similar sample pairs within the same embedding space while effectively separating dissimilar sample pairs. For example, InfoNEC [30] leverages information entropy to learn discriminative

representations from unlabeled data. SimCLR [5] learns representations by maximizing the correlation between representations of augmented views of the same image. MoCo [18] uses a momentum encoder to learn representations that are invariant to transformations. Yuan *et al.* [52] utilize inherent data attributes within each modality and cross-modal semantic information to construct cross-modal contrastive learning. Although contrastive learning has been explored in MLIR tasks [4, 17, 29, 47], to our knowledge, this is the first attempt to introduce multimodal contrastive learning into this task with missing labels.

## 3 Method

### 3.1 Problem Definition and Model Overview

In the multi-label image recognition with missing labels (MLR-ML) task, an image in a dataset of $C$ categories can be positive, negative, or unknown, with corresponding labels of 1, −1, and 0, respectively. Similar to prior work [16], we aim to leverage prompt tuning to learn a set of text representations that are functionally equivalent to classifiers.

As illustrated in Figure 2, to be specific, given an image $x$, the CLIP [33] visual encoder $\mathcal{F}(\cdot)$ maps it to a visual representation $f \in \mathbb{R}^{h \times w \times d}$, also called feature maps, where $w$, $h$, $d$ are width, height and dimension respectively. Note that, visual regions $R = \{r_1, r_2, \ldots, r_c\}$ indicate salient information about the different categories in an image or its feature maps. For the $c$-th category, its visual region $r_c = (p_c, e_c)$ contains the set of pixel indices $p_c$ and their corresponding energies $e_c$, where each element $p = (i, j)$ of $p_c$ represents a spatial position on the feature map and its corresponding energy $e \in e_c$ represents the probability of that pixel belonging to the $c$-th category. Then, we can leverage these regions to obtain region-level representation $F = \{f_1, f_2, \ldots, f_c\}$ for each category. To achieve one-to-one matching between text and images, we define text prompts for each category: $t_c = \{\omega_1, \omega_2, \ldots, \omega_L, \text{CLS}_c\}$, where $\omega_i$ is learnable prompt embedding and $\text{CLS}_c$ is class token for $c$-th category. Then, the text encoder $\mathcal{G}(\cdot)$ takes all $\{t_1, t_2, \ldots, t_c\}$ as input to generate text representations $G = \{g_1, g_2, \ldots, g_c\}$. Finally, the prediction score belonging to the $c$-th category is calculated as follows:

$$p\left(y_c \mid f_c, g_c\right) = \text{Matching}\left(f_c, g_c\right) = \frac{f_c \cdot g_c}{\|f_c\|_2 \|g_c\|_2} / \tau, \quad (1)$$

where $\tau$ is a learnable parameter, $\text{Matching}(\cdot)$ denotes $\tau$-normalized cosine similarity.

### 3.2 Category-Aware Region Learning

Visual region construction plays a key role in our text-region matching framework. A reliable visual region ensures that all pixels of a category are collected, resulting in accurate matching between the category-aware text representation and the region-level representation. Consequently, prompt tuning could be optimized to learn category-aware knowledge guided by accurate visual regions. The naïve way to construct a visual region $R$ is to use a binary mask to indicate the absence or presence of the category $c$ in the feature maps, *i.e.*, $r_c = \{r_{(i,j)} | i \in [1, h], j \in [1, w], r_{(i,j)} \in \{0, 1\}\}$. However, this is infeasible because we do not have access to pixel-level labels. To remedy this, we extend the naïve region from hard

form to soft form, which includes not only spatial positions but also present probabilities corresponding to these positions. The $r_c$, which performs the easy-to-hard form, can be redefined as follows:

$$r_c = (p_c, \text{sigmoid}(e_c)) , \quad (2)$$

where $p_c$ and $e_c$ consist of related positions and their present probability (energy), respectively. Specifically, we utilize CLIP text encoder with prompt tuning [55] or random initialization [3] to generate learnable category embeddings $Q = \{q_1, q_2, \ldots, q_c\}$. Next, we perform cross-attention using a decoder like a transformer-decoder [29], with category embeddings $Q$ as queries and visual representations $f$ as keys and values. This produces visual region $R$ and query-level representations $F^q = \{f_1^q, f_2^q, \ldots, f_c^q\}$. The above process is written as follows:

$$R = \text{Cross\_Att}(Q, f) ,$$
$$F^q = \text{Cross\_Att}(Q, f) + \text{MLP}(\text{Cross\_Att}(Q, f)) , \quad (3)$$

where $R$ is directly generated by Cross_Att. However, the transformer decoder contains multiple linear layers, which could disrupt the alignment of visual and textual representations in the joint space of CLIP. To address this problem, we leverage visual regions $R$ as soft mask to directly guide image representations $f$ to filter out irrelevant visual information and obtain region-level representations $F^r = \{f_1^r, f_2^r, \ldots, f_c^r\}$:

$$f_c^r = \sum_{(p_c, e_c) \in R} e_c \cdot f , \quad (4)$$

where $e_c$ is the energies of category $c$ in the visual representation at spatial location.

## 3.3 Knowledge Distillation for Matching

Processing high-quality visual regions plays a pivotal role in acquiring region-level representations. However, due to incomplete category annotation or missing pixel-level annotation, the visual regions learned by the transformer-decoder are often of poor quality (e.g., unable to locate objects, containing too much noise). This phenomenon could cause the two modalities between the text and visual to be unable to ensure good alignment. Thanks to CLIP's rich prior knowledge and generalization capabilities, we can transfer region-level representation $F^r$ knowledge to query-level representation $F^q$ to obtain higher-quality visual region $R$.

To be specific, according to Eq.(1), the query-level region representations $F^q$ and region-level representations $F^r$ are matched with the corresponding text representation $G$ to obtain prediction scores:

$$p_c^q \left(y_c \mid f_c^q, g_c\right) = \text{Matching}(f_c^q, g_c) ,$$
$$p_c^r \left(y_c \mid f_c^r, g_c\right) = \text{Matching}(f_c^r, g_c) . \quad (5)$$

Subsequently, we introduce knowledge distillation [20] to promote consistency between the above two types of representations, $F^r$ and $F^q$. This procedure can be expressed as follows:

$$\mathcal{L}_{\text{KD}} = \text{KL}(p^r || p^q) = \sum_i^C p_i^r \log \frac{p_i^r}{p_i^q} , \quad (6)$$

where KL is Kullback–Leibler divergence [20], $p_r$ and $p_q$ serve as the teacher prediction and student prediction, respectively.

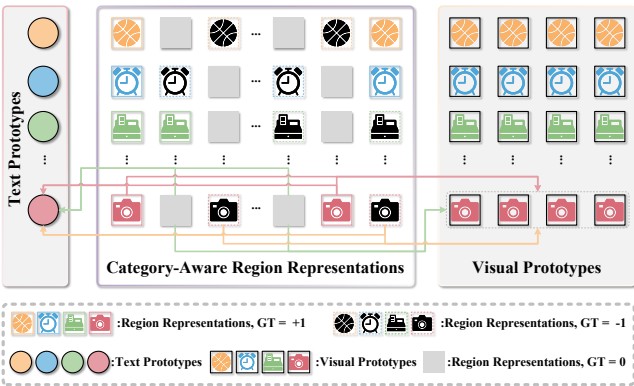

Figure 3: An overview of the pseudo-label generation process. To illustrate pseudo-label estimation simply and clearly, we select one of the four color samples as an example.

## 3.4 Multimodal Category Prototype

Intuitively, in a large-scale visual language pre-trained model (VLP), image-level, pixel-level, and text-level representations belonging to the same category are similar in terms of encoded features. Inspired by this, we hypothesize that region-level representations from the same category are highly similar to the corresponding visual or text prototype, and region-level representations from different images of the same category are also highly similar in the embedding space. To this end, we design a multimodal category prototype that includes visual and textual modalities.

**Visual Prototype.** To make the visual prototype more discriminative, we derive it from multiple pixel-level representations with high-energy rather than region-level representations. Specifically, we design a visual memory bank of size $c \times \ell \times d$ to store the pixel-level visual representations as visual prototypes. The visual repository consists of $c$ queues with a queue length of $\ell$. Given visual regions of an image, if the label of category $c$ is 1, we extract the top-k high response values on the $c$-th visual region as indexes and extract the representation of the corresponding position as visual prototypes to store them in the memory bank:

$$V_c = TopK(r_c) \cdot f , \quad (7)$$

where $V_c$ denotes the visual prototype of the $c$-th category. Then, $V_c$ is input to the $c$-th queue of the memory bank, using a first-in-first-out (FIFO) rule for the columns.

**Text Prototype.** In contrast to [22], we opt not to employ encoded representations of GPT-3 generated text descriptions as text prototypes [2]. This approach necessitates additional manual design efforts. Therefore, we adopt the text representation trained in the first stage as a text category prototype $T = \{t_1, t_2, \ldots, t_c\}$ to simplify and improve acquisition efficiency.

**Pseudo-Label Estimation.** Given a minibatch of input images $X = \{(x_i, y_i)\}_{i=1}^n$, we can obtain a set of region-level representations $F^r = \{f_{i,j}^r \mid i \in [1, n], \ j \in [1, c]\}$ using category-aware region learning, as described in the Sec. 3.2. The cosine similarity between the region-level representation $f^r$ and the visual prototype $v$ is

formulated as follows:

$$s = cosine\left(f^r, v\right) = \frac{f^r \cdot v}{\|f^r\|_2 \|v\|_2} \ . \tag{8}$$

As shown in Figure 3, we calculate the average cosine similarity between all region-level representations $f_c$ for the $c$-th category and the corresponding all visual prototypes $v_c$:

$$
\begin{aligned}
\bar{s}_c^p &= \frac{1}{|\mathcal{D}_c^p|} \sum_{m \in \mathcal{D}_c^p} s_{i,j}^c \cdot y_i^c, \quad \text{s.t. } i \in [1, n], j \in [1, \ell] , \\
\bar{s}_c^n &= \frac{-1}{|\mathcal{D}_c^n|} \sum_{m \in \mathcal{D}_c^n} s_{i,j}^c \cdot y_i^c, \quad \text{s.t. } i \in [1, n], j \in [1, \ell] ,
\end{aligned}
\tag{9}
$$

where $\bar{s}_c^p$ signifies the average similarity associated with positive labels, $\bar{s}_c^n$ denotes the average similarity linked to negative labels, $\mathcal{D}_c^p$ and $\mathcal{D}_c^n$ represent the positive label set and the negative label set, respectively. To estimate the label of a missing category $c$ in an image $m$, we calculate the average similarity between the visual prototype $v_c$ and the region representations $f_{m,c}$, and then compare it to two thresholds $\theta_{\text{positive}}^c$ and $\theta_{\text{negative}}^c$:

$$
\tilde{y}_{c;v}^n = \begin{cases}
1 & \bar{s}_c^u \geq \theta_{\text{positive}}^c, \ \bar{s}_c^u \leq \theta_{\text{negative}}^c \\
-1 & \bar{s}_c^u \geq \theta_{\text{negative}}^c, \ \bar{s}_c^u \leq \theta_{\text{positive}}^c , \\
0 & \text{otherwise}
\end{cases}
\tag{10}
$$

$$\bar{s}_c^u = \frac{1}{|\mathcal{D}_c^u|} \sum_{m \in \mathcal{D}_c^u} s_{i,j}^c, \quad \text{s.t. } i \in [m], j \in [1, \ell] ,$$

where $\mathcal{D}_c^u$ denotes the unknown label set. Moreover, the thresholds for different categories are dynamically and adaptively updated using an exponential moving average (EMA):

$$
\begin{aligned}
\theta_{\text{positive}}^c &\leftarrow \eta \theta_{\text{positive}}^c + (1 - \eta)\bar{s}_c^p , \\
\theta_{\text{negative}}^c &\leftarrow \eta \theta_{\text{negative}}^c + (1 - \eta)\bar{s}_c^n ,
\end{aligned}
\tag{11}
$$

where $\eta$ is the moving average parameter. On the other hand, we adopt a similar strategy to estimate unknown labels $\tilde{y}_{c;t}^n$ based on text prototypes $T$. Finally, the pseudo-labels estimated from visual prototypes and textual prototypes are combined:

$$
\tilde{y}_c^n = \begin{cases}
1 & \tilde{y}_{c;v}^n = 1, \ \tilde{y}_{c;t}^n = 1 \\
-1 & \tilde{y}_{c;v}^n = -1, \ \tilde{y}_{c;t}^n = -1 \ . \\
0 & \text{otherwise}
\end{cases}
\tag{12}
$$

## 3.5 Multimodal Contrastive Learning

To enhance the consistency between text and visual modalities and establish intra- and inter-class relationships, we design multimodal contrastive learning to bridge the semantic gap between modalities, enhance intra-class similarity, and diminish inter-class similarity. Specifically, we use the text representation of the $c$-th category as the anchor, and we select the region-level representation belonging to the same category from the current batch and memory bank as the positive sample $P$ while collecting the region-level representations from other categories as the negative samples $N$. We adopt the InfoNCE loss [30] as the contrastive loss:

$$\mathcal{L}_{\text{NCE}} = \frac{1}{|P|} \sum_{q_+ \in P} -\log \frac{\exp(p_a \cdot q_+ / \tau)}{\exp(p_a \cdot q_+ / \tau) + \sum_{q_- \in N} \exp(p_a \cdot q_- / \tau)}, \tag{13}$$

where $\tau$ is a temperature term that control the sharpness of the output. In addition, to strengthen semantic relationships (within and across categories) and minimize semantic gaps, we implement an additional multilayer perceptron (MLP) to enhance representation.

## 3.6 Learning & Inference

**Learning.** During the training phase, we apply classification losses to both the text-query matching prediction score and text-region matching prediction score, respectively: $\mathcal{L}_{cls} = \mathcal{L}_{cls}^q + \alpha \mathcal{L}_{cls}^r$. Following prior work [36], we utilize Asymmetric Loss (ASL) [34] as the classification loss optimization function:

$$
\mathcal{L}_{\text{cls}} = \begin{cases}
(1 - p)^{\gamma_+} \log(p), & y_k = 1, \\
(p_m)^{\gamma_-} \log(1 - p_m), & y_k = -1,
\end{cases}
\tag{14}
$$

where $p_m = \max(p - c, 0)$ represents the shifted probability for very easy negative samples. $\gamma_+$, $\gamma_-$, and $c$ values are set to 1, 2, and 0.05, respectively, by default. By combining classification loss with multimodal contrastive learning and knowledge distillation loss, we obtain the total loss function as follows:

$$\mathcal{L} = \mathcal{L}_{\text{cls}} + \beta \mathcal{L}_{\text{KD}} + \gamma \mathcal{L}_{\text{NCE}}, \tag{15}$$

where $\beta$ and $\gamma$ are hyperparameters.

**Inference.** For inference, we utilize the text-region matching score as the final prediction score, obtained after eliminating the multi-modal category prototype and loss function from our proposed model. It is worth noting that through Eq.(1), the prediction score obtained by our method is better than that obtained by pixel-level matching and image-level matching. Due to the limited space, we provide a detailed analysis of this conclusion in the Appendix.

## 4 Experiment

### 4.1 Experimental Setup

**Datasets & Evaluation.** To evaluate the effectiveness of our proposed method under the partial labels setting on the MLR-ML task, we conduct experiments on three benchmarks, MS-COCO [26], VOC 2007 [14] and Visual Genome [24], following the approach of SARB [32]. As the most widely used multi-label image recognition benchmark, MS-COCO contains 80 common categories, 82,081 images for training, and 40,137 images for testing. The VOC 2007 dataset consists of 9,963 images, split into 5,011 training images and 4,952 test images. The images are labeled with 20 object classes. Visual Genome (VG) is a large-scale image dataset with over 100,000 images, but most categories have few samples. Therefore, we select the 200 most frequent categories as a subset, also called VG-200, following the settings of SARB [32]. These datasets are fully annotated, so we randomly sample labels at a ratio of 10% to 90%, including both positive and negative labels. Furthermore, we perform validation to evaluate the effectiveness of the proposed method under the single positive label setting across four datasets, which include MS-COCO, NUS-WIDE [9], VOC 2012 [14], and CUB-200-2011 [37]. NUS-WIDE is a substantial web-based dataset frequently employed in multi-label image learning tasks, featuring a total of 81 categories. VOC 2012 is an extension of VOC 2007 with more images. The CUB-200-2011, a widely recognized benchmark dataset in fine-grained image recognition with 200 categories and 312 attributes, is also commonly used for multi-label image recognition tasks. Following

**Table 1: Compare our method with SOTA methods on the MS-COCO, VOC2007, and VG-200 benchmark datasets with partial labels. ★ indicates results we reproduced. Bolded means best performance, and all metrics are in %. All models adopt ResNet101 as the backbone. The "Vanilla" indicates weights pre-trained on ImageNet, while the "CLIP" utilizes weights derived from CLIP.**

| Methods | Venue | Backbone | 10% | 20% | 30% | 40% | 50% | 60% | 70% | 80% | 90% | Avg. |
|---|---|---|---|---|---|---|---|---|---|---|---|---|
| | | | | | | MS-COCO | | | | | | |
| Curriculum labeling [13] | CVPR'19 | Vanilla | 26.7 | 31.8 | 51.5 | 65.4 | 70.0 | 71.9 | 74.0 | 77.4 | 78.0 | 60.7 |
| Patial-BCE [13] | CVPR'19 | Vanilla | 61.6 | 70.5 | 74.1 | 76.3 | 77.2 | 77.7 | 78.2 | 78.4 | 78.5 | 74.7 |
| SST [6] | AAAI'22 | Vanilla | 68.1 | 73.5 | 75.9 | 77.3 | 78.1 | 78.9 | 79.2 | 79.6 | 79.9 | 76.7 |
| SARB [32] | AAAI'22 | Vanilla | 71.2 | 75.0 | 77.1 | 78.3 | 78.9 | 79.6 | 79.8 | 80.5 | 80.5 | 77.9 |
| DualCoOp [36] | NeurIPS'22 | CLIP | 78.7 | 80.9 | 81.7 | 82.0 | 82.5 | 82.7 | 82.8 | 83.0 | 83.1 | 81.9 |
| DualCoOp★ [36] | NeurIPS'22 | CLIP | 81.5 | 82.7 | 83.3 | 83.8 | 84.0 | 84.2 | 84.4 | 84.4 | 84.5 | 83.6 |
| TaI-DPT+DualCoOp [16] | CVPR'23 | CLIP | 81.5 | 82.6 | 83.3 | 83.7 | 83.9 | 84.0 | 84.2 | 84.4 | 84.5 | 83.6 |
| SCPNet [12] | CVPR'23 | CLIP | 80.3 | 82.2 | 82.8 | 83.4 | 83.8 | 83.9 | 84.0 | 84.1 | 84.2 | 83.2 |
| HSPNet [38] | MM'23 | CLIP | 78.3 | 81.4 | 82.2 | 83.6 | 84.3 | 84.8 | 85.0 | 85.4 | 85.6 | 83.4 |
| TaI-Adapter+DualCoOp [56] | ArXiv'23 | CLIP | 82.1 | 82.9 | 83.5 | 84.0 | 84.4 | 84.7 | 84.9 | 85.1 | 85.1 | 84.1 |
| TRM-ML | Ours | CLIP | **83.3** | **84.5** | **85.0** | **85.3** | **85.6** | **85.8** | **86.1** | **86.4** | **86.5** | **85.4** |
| | | | | | | PASCAL VOC 2007 | | | | | | |
| Patial-BCE [13] | CVPR'19 | Vanilla | 80.7 | 88.4 | 89.9 | 90.7 | 91.2 | 91.8 | 92.3 | 92.4 | 92.5 | 90.0 |
| SST [6] | AAAI'22 | Vanilla | 81.5 | 89.0 | 90.3 | 91.0 | 91.6 | 92.0 | 92.5 | 92.6 | 92.7 | 90.4 |
| SARB [32] | AAAI'22 | Vanilla | 83.5 | 88.6 | 90.7 | 91.4 | 91.9 | 92.2 | 92.6 | 92.8 | 92.9 | 90.7 |
| DualCoOp [36] | NeurIPS'22 | CLIP | 90.3 | 92.2 | 92.8 | 93.3 | 93.6 | 93.9 | 94.0 | 94.1 | 94.2 | 93.2 |
| DualCoOp★ [36] | NeurIPS'22 | CLIP | 91.8 | 93.3 | 93.7 | 94.2 | 94.2 | 94.7 | 94.8 | 94.8 | 94.9 | 94.0 |
| TaI-DPT+DualCoOp [16] | CVPR'23 | CLIP | 93.3 | 94.6 | 94.8 | 94.9 | 95.1 | 95.0 | 95.1 | 95.3 | 95.5 | 94.8 |
| SCPNet [12] | CVPR'23 | CLIP | 91.1 | 92.8 | 93.5 | 93.6 | 93.8 | 94.0 | 94.1 | 94.2 | 94.3 | 93.5 |
| TaI-Adapter+DualCoOp [56] | ArXiv'23 | CLIP | 93.8 | 94.7 | 95.1 | 95.2 | 95.3 | 95.3 | 95.4 | 95.6 | 95.7 | 95.1 |
| TRM-ML | Ours | CLIP | 93.9 | 94.6 | 94.9 | 95.3 | 95.4 | 95.6 | 95.6 | 95.6 | 95.7 | 95.2 |
| TaI-DPT+TRM-ML | Ours | CLIP | **94.5** | **95.0** | **95.3** | **95.4** | **95.6** | **95.7** | **95.7** | **95.8** | **95.9** | **95.4** |
| | | | | | | VG-200 | | | | | | |
| SSGRL [7] | ICCV'19 | Vanilla | 34.6 | 37.3 | 39.2 | 40.1 | 40.4 | 41.0 | 41.3 | 41.6 | 42.1 | 39.7 |
| ML-GCN [8] | CVPR'19 | Vanilla | 32.0 | 37.8 | 38.8 | 39.1 | 39.6 | 40.0 | 41.9 | 42.3 | 42.5 | 39.3 |
| SST [6] | AAAI'22 | Vanilla | 38.8 | 39.4 | 41.1 | 41.8 | 42.7 | 42.9 | 43.0 | 43.2 | 43.5 | 41.8 |
| SARB [32] | AAAI'22 | Vanilla | 41.4 | 44.0 | 44.8 | 45.5 | 46.6 | 47.5 | 47.8 | 48.0 | 48.2 | 46.0 |
| SCPNet [12] | CVPR'23 | CLIP | 43.8 | 46.4 | 48.2 | 49.6 | 50.4 | 50.9 | 51.3 | 51.6 | 52.0 | 49.4 |
| TRM-ML | Ours | CLIP | **50.5** | **51.9** | **52.0** | **53.0** | **53.3** | **53.5** | **53.6** | **53.7** | **53.9** | **52.8** |

prior researchs [12, 16, 32], we employ mean Average Precision (mAP) as the primary evaluation metric and provide an average mAP score for a comprehensive assessment.

**Implementation details.** To facilitate a fair comparison, we employ ResNet [19] as the visual encoder, and Transformer as the text encoder, both initialized with parameters from CLIP pre-training [33], also known as CLIP ResNet. The weights for vanilla ResNet are obtained from PyTorch [31]. The class-specific text prompts adopt the dual semantic strategy, initialized with Gaussian noise sampled from $\mathcal{N}(0, 0.02)$, and the length of the prompts is 16. Note that both the visual encoder and text encoder are frozen during training phase. The transformer decoder has only one attention head. We use AdamW as the optimizer and OneCycleLR as the lr_scheduler, with a maximum learning rate of 5e-4 and a batch size of 128. The input resolution of all images is $448 \times 448$, and the data augmentation settings are consistent with TaI-DPT [16]. We train for 40 epochs on all benchmark datasets. The default values for the hyperparameters are $\alpha = 1$, $\beta = 0.05$, and $\gamma = 0.01$.

## 4.2 Comparisons with SOTA Methods

**MLR-ML on partial labels setting.** To evaluate the effectiveness of our proposed method for the multi-label image recognition with partial labels task, we compare it to three types of methods: conventional (*e.g.*, Curriculum labeling [13], Patial-BCE [13]), graph-based (*e.g.*, ML-GCN [8], SSGRL [7], SST [6], SARB [32]), and prompt tuning (*e.g.*, DualCoOp [36], TaI-DPT [16], SCPNet [12], HSPNet [38], TaI-Adapter [56]). Although prompt tuning methods adopt CLIP ResNet101 as their visual encoder, which is better than other methods, previous work, *e.g.*, TaI-DPT, has confirmed that most prompt tuning methods are superior to the other two types of methods. As shown in Table 1, our proposed method outperforms existing methods on all benchmark datasets by a large margin. The proposed method demonstrates superior performance compared to HSPNet, SCPNet, and TaI-DPT on the MS-COCO dataset, surpassing them by 2.0%, 2.2%, and 1.3%, respectively. Although it outperforms TaI-DPT by 0.4% on VOC 2007, a combined approach incorporating both methods results in a notable 0.6% enhancement.

Table 2: Comparison with the SOTA methods for MLR with single positive label. In this table, VOC refers to the VOC 2012 benchmark dataset. ⋆ means freeze visual encoder. All models adopt ResNet50 as the backbone. The "Vanilla" indicates weights pre-trained on ImageNet, while the "CLIP" utilizes weights derived from CLIP. Bolded means best performance.

| Method | Venue | Backbone | LargeLoss setup [23] | | | | | SPLC setup [53] | | | | |
|--------|-------|----------|------|-----|-----|-----|------|------|-----|-----|-----|------|
| | | | COCO | VOC | NUS | CUB | Avg. | COCO | VOC | NUS | CUB | Avg. |
| LSAN [10] | CVPR'21 | Vanilla | 69.2 | 86.7 | 50.5 | 17.9 | 56.1 | 70.5 | 87.2 | 52.5 | 18.9 | 57.3 |
| ROLE [10] | CVPR'21 | Vanilla | 69.0 | 88.2 | 51.0 | 16.8 | 56.3 | 70.9 | 89.0 | 50.6 | 20.4 | 57.7 |
| LargeLoss [23] | CVPR'22 | Vanilla | 71.6 | 89.3 | 49.6 | 21.8 | 58.1 | - | - | - | - | - |
| Hill [53] | arXiv'21 | Vanilla | - | - | - | - | - | 73.2 | 87.8 | 55.0 | 18.8 | 58.7 |
| SPLC [53] | arXiv'21 | Vanilla | 72.0 | 87.7 | 49.8 | 18.0 | 56.9 | 73.2 | 88.1 | 55.2 | 20.0 | 59.1 |
| HSPNet [38] | MM'23 | CLIP | 74.8 | 89.4 | 56.3 | 23.4 | 61.0 | 75.7 | 90.4 | 61.8 | 24.3 | 63.1 |
| SCPNet [12] | CVPR'23 | CLIP | 75.4 | 90.1 | 55.7 | 25.4 | 61.7 | 76.4 | 91.2 | 62.0 | 25.7 | 63.8 |
| TRM-ML⋆ | Ours | CLIP | **78.6** | **91.8** | **57.3** | **26.9** | **63.7** | **79.2** | **92.2** | **63.4** | **27.6** | **65.6** |

Remarkably, TaI-DPT or TaI-Adapter combined with DualCoOp surpasses most methods on VOC 2007, especially in settings with a low proportion of known labels. This difference arises from the substantial disparity in both the volume and completeness of training data between text and image. Text data typically comprises a larger dataset with more comprehensive annotations compared to image data. Furthermore, when evaluated on the more demanding VG-200 dataset, the proposed method outperforms the suboptimal method by a notable margin of 3.4%.

**MLR-ML on single positive labels setting.** To fairly evaluate the effectiveness of our proposed method for the multi-label image recognition with single positive labels task, we adopt the experimental settings of LargeLoss [23] and SPLC [53]. Table 2 clearly demonstrates that our proposed method surpasses existing methods across all four benchmark datasets. Remarkably, even when employing a frozen CLIP ResNet50 as the visual encoder, our method outperforms HSPNet and SCPNet, exhibiting substantial improvements, with 1.7% and 2.4% enhancement on the VOC 2012 dataset and impressive 3.8% and 3.2% increase on the MS-COCO dataset, particularly under the LargeLoss setup. Overall, the performance under the SPLC setup is 1.9% superior to the LargeLoss setup. However, compared to SCPNet, our method achieves a 2.0% improvement under the LargeLoss setting and a 1.8% improvement under the SPLC setup.

## 4.3 Ablation study

In this section, to better understand how our method improves multi-label image recognition with missing labels, we conduct a series of experiments on the VOC 2007 and MS-COCO datasets.

**The effect of key components.** To investigate the impact of different components on the model's performance and validate its effectiveness, we conduct an ablation study, systematically removing or modifying different components and evaluating the resulting model's performance. In this work, we employ the modified DualCoOp as a baseline, retaining the dual semantic prompt, text encoder, and visual encoder upon integrating our proposed component and omitting the attention module. As shown in Table 3, our baseline method outperforms the SATO method, and integrating our proposed components with the baseline method steadily improves performance compared to the baseline alone. This demonstrates

Table 3: Ablation study of different components in VOC 2007 and MS-COCO datasets with partial labels. CARL stands for category-aware region learning. KD refers to the use of knowledge distillation. MMCP represents pseudo-label estimation supported by multimodal category prototypes. MMCL denotes multimodal contrastive learning. The retention rate of labels varies from 10% to 90%. Bolded means best performance, and all metrics are in %.

| Components | | | | | MS-COCO | VOC 2007 | Avg. |
|-----------|------|----|------|------|---------|----------|------|
| Baseline | CARL | KD | MMCP | MMCL | | | |
| ✓ | ✗ | ✗ | ✗ | ✗ | 77.7 | 90.8 | 84.3 |
| ✓ | ✓ | ✗ | ✗ | ✗ | 84.1 | 94.2 | 89.2 |
| ✓ | ✓ | ✓ | ✗ | ✗ | 84.4 | 94.6 | 89.5 |
| ✓ | ✓ | ✓ | ✓ | ✗ | 84.7 | 94.9 | 89.8 |
| ✓ | ✓ | ✓ | ✗ | ✓ | 85.0 | 94.7 | 89.9 |
| ✓ | ✓ | ✓ | ✓ | ✓ | **85.4** | **95.2** | **90.3** |

the effectiveness and significance of our proposed components in addressing missing label tasks through prompt tuning.

**The effect of visual regions.** We further evaluate the impact of high-quality visual regions on text prompt tuning. To this end, we remove multimodal category prototypes and multimodal contrastive learning, train the model under complete annotation, and load the trained category-aware region learning module into a new model. Finally, we train the new model with partial label data. As presented in Table 4, we report mAP for 10%-50% annotation proportions, with the average mAP shown in the last column. Leveraging high-quality visual regions, we achieve an average mAP improvement of 1.0% and 0.8% on VOC 2007 and MS-COCO datasets, respectively. The most notable enhancements are observed at 10% and 20% annotation rates, with mAP gains of 2.8% and 1.0%, respectively, for VOC 2007, and 2.3% and 0.9%, respectively, for MS-COCO. These results demonstrate the effectiveness of high-quality visual regions in optimizing text prompt tuning.

## 4.4 Model Analysis

**Number of parameters and computational overhead.** Described in Table 5, evaluated on MS-COCO, these methods have

**Table 4: Ablation study examines the influence of high-quality visual regions on text prompt tuning learning in the MS-COCO and VOC 2007 datasets. The "complete" refers to the category-aware region learning module, trained using complete annotation data. All metrics are in %.**

| Dataset | Method | 10% | 20% | 30% | 40% | 50% | Avg. |
|---------|--------|-----|-----|-----|-----|-----|------|
| VOC 2007 | w/o complete | 92.0 | 93.9 | 94.4 | 94.9 | 95.0 | 94.0 |
| | w complete | 94.8 | 94.9 | 95.0 | 95.1 | 95.1 | 95.0 |
| MS-COCO | w/o complete | 80.8 | 82.9 | 83.7 | 84.3 | 84.6 | 83.3 |
| | w complete | 83.1 | 83.8 | 84.2 | 84.5 | 84.7 | 84.1 |

**Table 5: Computation cost in inference and parameters number on MS-COCO.**

| Method | # Total param. | # Learnable param. | FLOPs | mAP |
|--------|----------------|---------------------|-------|-----|
| DualCoOp | 91.17 M | 1.25 M | 36.97 G | 81.9% |
| SCPNet | 96.02 M | 59.66 M | 36.60 G | 83.2% |
| TRM-ML(ours) | 96.72 M | 6.80 M | 37.50 G | 85.4% |

**Table 6: Text prototype and visual prototype valid independently. The retention rate of labels varies from 10% to 50%. "Text" is text prototype, and "Visual" means visual prototype.**

| Components | | | | | VOC 2007 |
|------------|------|----|------|--------|----------|
| Baseline | CARL | KD | Text | Visual | |
| ✓ | ✓ | ✓ | ✗ | ✗ | 93.24 |
| ✓ | ✓ | ✓ | ✓ | ✗ | 94.29 |
| ✓ | ✓ | ✓ | ✗ | ✓ | 94.15 |
| ✓ | ✓ | ✓ | ✓ | ✓ | **94.76** |

comparable total parameters and FLOPS, with little difference. TRM-ML has slightly more learnable parameters than DualCoOp, but significantly less than SCPNet. We achieve a higher mAP than both. In general, our method is superior to the other two methods.

**Text prototype and visual prototype valid independently.** We conduct experiments on the VOC 2007. Notably, multimodal contrastive learning has been removed. In Table 6, text prototypes and visual prototypes significantly improve model performance. In isolation, text prototypes achieve a 1.05% improvement, while visual prototypes achieve a 0.91% improvement. Combining both prototypes yields an even greater improvement of 1.52%.

### 4.5 Visualization

To demonstrate the effectiveness of category-aware region learning in constructing visual regions associated with specific categories, we present a comparison between the visual regions generated by the presented method and the class activation maps corresponding to DualCoOp, along with the corresponding prediction scores, as depicted in Figure 4. From the figure, the four common scene images in MS-COCO are shown: *outdoor*, *food*, *sports*, and *transportation*. 1) Compared to DualCoOp, the proposed method excels in extracting visual information from category-specific regions

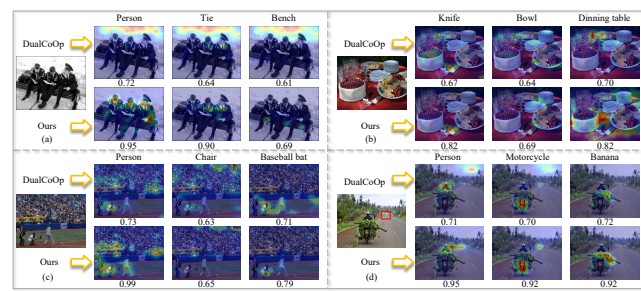

**Figure 4: Visual analysis of DualCoOp and the proposed method. For each subfigure, we present the corresponding categories of the top-3 prediction scores, with the first row from DualCoOp and the second row generated by category-aware region learning module.**

while mitigating the impact of noise, as demonstrated by the examples of "*person*" in Figure 4(a) and "*motorcycle*" in Figure 4(d). 2) Our proposed method exhibits improved accuracy in perceiving certain objects, particularly smaller objects within complex scenes, *e.g.*, "*tie*" in Figure 4(a), "*knife*" in Figure 4(b) and "*person*" in Figure 4(c). Please observe that within the input image depicted in Figure 4(d), both a "*motorcycle*" and a "*person*" are delineated within the red box and our method can locate them. In cases where certain categories are present in an image, our proposed method yields higher prediction scores than DualCoOp. In summary, category-aware region learning allows for a more precise emphasis on relevant visual regions, leading to enhanced prediction accuracy.

## 5 Conclusion

In this work, to address the text-visual optimization challenges encountered by visual and language pre-trained models when dealing with multi-label image recognition with missing labels (MLR-ML) tasks, we introduce a novel framework called TRM-ML. To this end, we propose a category-aware region learning module that transforms the one-to-many (*pixel-level*) or one-to-agnostic (*image-level*) matching problem into a one-to-one (*region-level*) matching problem, enabling more effective multi-label image recognition. Furthermore, we combine multimodal contrastive learning with the category-aware region learning module to reduce the semantic gaps between textual and visual representations. When these three components are integrated, our method achieves new state-of-the-art performance on diverse multi-label benchmark datasets.

## Acknowledgments

This work was supported in part by the National Natural Science Foundation of China (Grant No. 62076005, U20A20398), the Natural Science Foundation of Anhui Province (Grant No. 2008085MF191, 2308085MF214), and the University Synergy Innovation Program of Anhui Province, China (Grant No. GXXT-2021-002, GXXT-2022-029). We thank Ming-Kun Xie (He is a postdoctoral researcher at RIKEN Center for Advanced Intelligence Project, in Japan) for his helpful discussions. We also thank the Hefei Artificial Intelligence Computing Center of Hefei Big Data Asset Operation Co., Ltd. for providing computational resources for this project.

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
