# OpenReview forum: "Text-Region Matching for Multi-Label Image Recognition with Missing Labels"
_acmmm.org/ACMMM/2024/Conference — MM2024 Poster_

### Official Review · Reviewer_WRAp · 2024-05-18

**Rating:** 3
**Confidence:** 3

**Summary:**

This paper introduces a novel framework called TRM-ML that addresses the challenge of missing labels in multi-label image recognition tasks. By integrating a Category-Aware Region Learning Module, Multimodal Contrastive Learning, and Multimodal Category Prototypes, the framework effectively bridges the semantic gap between textual and visual representations, achieving state-of-the-art performance on several multi-label benchmark datasets. The authors demonstrate the effectiveness and superiority of their approach through a series of ablation studies and visualization analyses, particularly in dealing with incomplete annotations.

**Strengths:**

1.The methodology is technically sound, with a clear explanation of how the text-region matching strategy works in conjunction with multimodal contrastive learning and the multimodal category prototype.
2.The authors have conducted extensive experiments on multiple benchmark datasets, which demonstrates the effectiveness of their proposed framework.
3.The manuscript is well-written, and the figures and illustrations are clear and aid in understanding the proposed method.

**Limitations:**

1.The proposed TRM-ML framework incorporates several new components, including a category-aware region learning module and a multimodal category prototype. However, the paper does not discuss the computational complexity of these additions, which could be significant, especially for large-scale datasets. An analysis of the computational overhead would be beneficial.

2.Including qualitative case studies or examples where the model succeeds or fails could provide deeper insights into the model's capabilities and limitations.

3.A discussion on the sensitivity to hyperparameter choices would be beneficial, as this can greatly affect the performance of models.

**Suitability:**

3

---

### Official Review · Reviewer_gecv · 2024-05-25

**Rating:** 4
**Confidence:** 3

**Summary:**

This paper introduces a novel method, Text-Region Matching (TRM-ML), for optimizing multi-label prompt tuning to improve multi-label image recognition with incomplete annotations. TRM-ML narrows down the semantic gap between text and vision by exploring the information of category-aware regions rather than the entire image or pixels. It incorporates multimodal contrastive learning and a multimodal category prototype to establish intra-class and inter-class relationships and estimate unknown labels, aiding pseudo-label generation.

**Strengths:**

1.The paper presents some focused and customized designs for the task of multi-label image recognition with incomplete annotations.

2.It conducts relatively comprehensive experiments, providing a robust validation of its proposed methods. The proposed methods clearly demonstrate significant improvement over existing strategies, showing the effectiveness of the proposed solutions.

**Limitations:**

1.While the authors claim to have four main innovative points, it would be helpful for them to condense and emphasize their core contribution, particularly focusing on the methodology outlined in Figure 2. At present, the work feels like a combination of several smaller methods, making it difficult to understand the systematized value of the proposed work.

2.The study could benefit from more comprehensive experiments. The authors could analyze the performance on larger category datasets like OpenImages to extend the applicability of their method beyond the current scope. These additional investigations would increase the paper’s relevance to real-world applications and widen its usability.

**Suitability:**

3

---

### Official Review · Reviewer_WaPw · 2024-06-06

**Rating:** 4
**Confidence:** 3

**Summary:**

The paper introduces Text-Region Matching for Multi-Label prompt tuning (TRM-ML), aiming to enhance cross-modal matching in multi-label image recognition with missing labels. It emphasizes category-aware regions, utilizes multimodal contrastive learning, and introduces a multimodal category prototype to estimate unknown labels. Experimental results demonstrate the superiority of TRM-ML over existing methods.

**Strengths:**

1）This paper is easy to understand, with a great introduction that thoroughly analyzes the shortcomings of existing work.
2）The related experiments are conducted comprehensively.
3）Proposed a relatively novel approach is proposed.

**Limitations:**

1）The authors pointed out in the introduction that text-to-pixel level matching may introduce irrelevant noise. However, in Section 3.4, they chose to use pixel-level representations to construct visual prototypes, which seems contradictory to their previous statement. Why not use existing region representations to build visual prototypes, which could also save computational time?
2）If high-energy pixels are used for filtering to address pixel noise issues, is it worthwhile to explore whether integrating high-energy pixels directly into DualCoop could achieve similar effects without adding extra computational burden?
3）In generating region representations, the cross-attention mechanism of the transformer-decoder is employed. However, considering the actual spatial information in the images might yield better results.such as SCLIP[1].
[1] SCLIP:Rethinking Self-Attention for Dense Vision-Language Inference.
4）The framework diagram contains too many lines, which can lead to confusion. For example, while generating visual prototypes, the text mentions using pixel-level representations, but the diagram shows lines from Fr​ pointing to visual prototypes, which seems incorrect.
Minor Issues:
1)Table 6 should indicate that the last column represents the mean retention rate of labels varying from 10% to 50%.
2)There are minor errors, such as "SATO" on line 842 of page 8, which should be "SOTA".

**Suitability:**

3

---

### Official Review · Reviewer_7oeJ · 2024-06-07

**Rating:** 4
**Confidence:** 2

**Summary:**

This paper proposes a text-region matching for multi-label image recognition. The experiments prove the effectiveness of the proposed  method.

**Strengths:**

The paper presents a one-to-one (category-to-region) matching between textual and visual representations.

The proposed TRM-ML reaches the SOTA performance.

**Limitations:**

The labels are randomly sampled for training, according to sec4.1.The paper should include the standard deviation in the performance of TRM-ML when using different seeds.

Why the baseline of TRM-ML surpasses some SOTA methods?

The paper should include the analysis on the influence of 𝜂 (moving average parameter).

**Suitability:**

3

---

### Meta-Review · Area_Chair_qvcW · 2024-06-28

**Recommendation:** Accept (Poster)
**Confidence:** 4

**Metareview:**

Initially, the paper received three borderline Accept scores and one borderline Reject score. The reviewers liked the proposed approach to multi-label image recognition with missing labels by exploring category-aware regions' information to narrow the semantic gap between text and vision. However, they were concerned about the lack of multi-run experiments, the rationale of methodology designs, and the lack of necessary analysis for model complexity and hyper-parameters. These issues made it difficult to evaluate the proposed method. The rebuttal addressed the reviewers’ points by offering additional experimental results and explanations.

After rebuttal, Reviewer 7oeJ and WaPw kept the borderline Accept scores, and Reviewer WRAp changed the rating from borderline Reject to borderline Accept.  They recommended acceptance, acknowledging that the rebuttal successfully addressed their concerns.

The AC has read the paper, reviews, and rebuttal. Reviewer gecv rated borderline Accept initially and didn't confirm with final recommendation. However, the AC believes the authors' response is reasonable, and Reviewer gecv's concerns will not overturn the overall rating. The authors are encouraged to improve the final paper version by following reviewer recommendations, in particular including a clearer ablation analysis and more in-depth discussion, as well as improving the clarity of the writing and presentation.